# Isolation and Characterization of Two Chalcone Derivatives with Anti-Hepatitis B Virus Activity from the Endemic Socotraen *Dracaena cinnabari* (Dragon’s Blood Tree)

**DOI:** 10.3390/molecules27030952

**Published:** 2022-01-30

**Authors:** Ramzi A. Mothana, Ahmed H. Arbab, Ali A. ElGamal, Mohammad K. Parvez, Mohammed S. Al-Dosari

**Affiliations:** Department of Pharmacognosy, College of Pharmacy, King Saud University, Riyadh 11451, Saudi Arabia; arbabssn@gmail.com (A.H.A.); aelgamel@ksu.edu.sa (A.A.E.); mohkhalid@ksu.edu.sa (M.K.P.); mdosari@ksu.edu.sa (M.S.A.-D.)

**Keywords:** hepatitis B, *Dracaena cinnabari*, socotra, chalcones, antiviral, Yemen

## Abstract

Hepatitis B virus (HBV) infection is prevalent and continues to be a global health concern. In this study, we determined the anti-hepatitis B virus (HBV) potential of the Socotra-endemic medicinal plant *Dracaena cinnabari* and isolated and characterized the responsible constituents. A bioassay-guided fractionation using different chromatographic techniques of the methanolic extract of *D. cinnabari* led to the isolation of two chalcone derivatives. Using a variety of spectroscopic techniques, including ^1^H-, ^13^C-, and 2D-NMR, these derivatives were identified as 2,4’-dihydroxy-4-methoxydihydrochalcone (compound **1**) and 2,4’-dihydroxy-4-methoxyhydrochalcone (compound **2**). Both compounds were isolated for the first time from the red resin (dragon’s blood) of *D. cinnabari*. The compounds were first evaluated for cytotoxicity on HepG2.2.15 cells and 50% cytotoxicity concentration (CC50) values were determined. They were then evaluated for anti-HBV activity against HepG2.2.15 cells by assessing the suppression of HBsAg and HBeAg production in the culture supernatants and their half maximum inhibitory concentration (IC_50_) and therapeutic index (TI) values were determined. Compounds **1** and **2** indicated inhibition of HBsAg production in a dose- and time-dependent manner with IC_50_ values of 20.56 and 6.36 μg/mL, respectively.

## 1. Introduction

Infectious disease caused by the hepatitis B virus (HBV) is widespread and continues to be a global health issue. Globally, around two billion people show evidence of infection with hepatitis, and about 250–360 million people are chronically infected and at risk for liver disease, according to statistics. Cirrhosis, liver failure, or hepatocellular carcinoma develop in 15–40% of people infected with HBV [1,2,3]. HBV is a DNA virus that encodes surface antigen (HBsAg), core antigen (HBcAg), and pre-core antigen (HBeAg), which are used as diagnostic markers of HBV infection [1,2].

Around 25% of chronically infected adults develop cirrhosis or liver cancer and die as a result of this infection [4,5,6]. All existing preventive and treatment approaches, such as HBV vaccines, interferons, and nucleotide, or nucleoside analogues (NAs; lamivudine, adefovir, entecavir, etc.) have their own limitations, especially due to the emergence of NA-associated drug resistance [7,8]. Furthermore, the price of these anti-HBV drugs makes them unaffordable for most underdeveloped countries. As a result, new antivirals, such as novel phytoproducts, need to be developed quickly and with increased strength and effectiveness. Continuous research is being conducted to discover novel anti-HBV medications derived from natural sources that have higher efficacy and promise. Several phytocompounds from distinct phytochemical classes have been studied for their potential anti-HBV activity [1,9,10,11,12].

The island of Socotra in the Arabian Sea constitutes a distinct part of Yemen. As a result of its biodiversity and endemism in a wide range of terrestrial and marine groups, including plants, it has global relevance for conservation of biodiversity. Among the 825 species recorded in Socotra, 307 (or 37%) are unique to the region [13]. *Dracaena cinnabari* Balf. f. (dragon’s blood tree, Asparagaceae) is one of the island’s most interesting endemic trees. For centuries, people have utilized the tree’s resin to treat anything from diarrhea and dysentery to ulcers [14,15]. Previous studies on the resin of *D. cinnabari* have shown interesting pharmacological activities including hemostasis, muscle relaxation, antidiabetic, anti-inflammatory, analgesic, antimicrobial, and anticancer effects [16,17,18,19]. Recently, we reported that the methanolic extract of *D. cinnabari* exhibited a promising anti-hepatitis B virus (HBV) potential [3]. In continuation of our work on this herbal drug, we studied the isolation and structure elucidation of two chalcone derivatives with in vitro anti-hepatitis B virus (HBV) activity.

## 2. Results

As part of a continuing study of medicinal plants and natural products with antiviral activity, we studied the isolation and structure elucidation of two chalcone derivatives from *Dracaena cinnabari* (dragon’s blood tree) with anti-hepatitis activity.

### 2.1. Structure Elucidation of the Isolated Compounds

A bioassay-guided fractionation, using chromatographic techniques, of the methanolic extract of *D. cinnabari* led to the isolation of two chalcone derivatives that showed anti-hepatitis activity. The isolated compounds were identified using a variety of spectroscopic techniques, including ^1^H-, ^13^C-, and 2D-NMR (COSY, HSQC, and HMBC). The NMR data of compounds **1** and **2** were comparable. The electron ionization mass spectra (EIMS) of compounds **1** and **2** revealed [M+] ions at *m*/*z* 272 and 270, corresponding to the chemical formulas C_16_H_16_O_4_ and C_16_H_14_O_4_, respectively. Sixteen carbon atoms were assigned to both compounds by ^13^C NMR and DEPT experiments and distinguished into two methylene, seven methine, one aromatic methoxy group, and six quaternaries for compound **1,** while two methines were substituted for the two methylenes for compound **2**. ^1^HNMR for compound **1** revealed the presence of two coupled methylene triplets at *δ* 3.10 and 2.85 connected directly to carbon at δ 40.0 and 26.9 ppm and assigned via the HSQC experiment. The remaining NMR proton signals were assigned to one aromatic methoxy group at 3.78 ppm and two sets of seven aromatic protons. One appeared as an ABX system at *δ* 6.41, 6.26, and 6.93 assigned to protons 3, 5, and 6, respectively. The second set of aromatic residues appeared at 6.83 and 7.87 ppm, each integrating two protons coupled to each other in the COSY experiment, which split into a doublet with a large coupling constant (*J* = 8.8 Hz, indicating the presence of four aromatic protons in a p-substituted aromatic ring. A downfield shift carbonyl carbon appeared at *δ* 201.8 and showed *J*2 and *J*3 bond cross-peak correlations with aromatic protons at 7.87 (2’,6,) ppm and with those at *δ* 2.85 and 3.10 (α and β) in the HMBC experiment. The data revealed that the structure of compound **1** is a dihydrochalcone derivative and its structure was assigned as 2,4’-dihydroxy-4-methoxydihydrochalcone according to HMBC correlations from methylene protons at *δ* 3.10 (2H, t) to C1 at d 121.7, from H6 at *δ* 6.93 to β carbon at 26.9 ppm, and from H6 to C4 at *δ* 162.5, and C2 at 161.3 ppm. Thus, the chemical structure of compound **1** was assigned as 2,4’-dihydroxy-4-methoxydihydrochalcone (Figure 1). The spectral data of compound **2** were similar to those of compound **1** except for the lack of two methylene protons at *δ* 3.10 and 2.85 (α and β); instead, the emergence of two trans olefinic protons resonated at *δ* 7.52 (1H, *J* = 15.7 Hz) and (7.57 1H, *J* = 15.7 Hz), connected directly though one bond length to carbons at *δ* 125.1 and 144.2 ppm in HSQC, which were assigned to positions α and β, respectively (Table 1). The rest of the spectral data were similar to those of compound **1** and indicated its chalcone nature. The structure of compound **2** was confirmed to be 2,4’-dihydroxy-4-methoxyhydrochalcone (Figure 1). Both compounds were isolated for the first time from the red resin of *D. cinnabari*.

### 2.2. Effects of the Isolated Compounds on Cell Viability

The cytotoxicity of compounds **1** and **2** was evaluated by determining the CC_50_ in the appropriate ranges of their toxic concentration. The CC_50_ values are demonstrated in Table 2. Compounds **1** and **2** showed extremely weak cytotoxicity with CC_50_ values of 346 and 242 μg/mL and a high therapeutic index (TI) of 16.8 and 38.0, respectively. The microscopic observation of cell morphology and growth after treatment with various concentrations of each compound (data not shown) supported this result. Consequently, both compounds at the non-cytotoxic concentration of 50 μg/mL were used in the subsequent studies of their antiviral effects.

### 2.3. Dose- and Time-Dependent Inhibition of Hepatitis B Virus Surface Antigen (HBsAg) Expression

Different non cytotoxic concentrations of compounds **1** and **2** were screened for anti-HBV activities by measuring the expression levels of viral HBsAg at day 3 and 5 after treatment. Of these, both compounds showed inhibition of HBsAg production in a dose- and time-dependent manner. The IC_50_ values of compounds **1** and **2** were estimated as 20.56 and 6.36 μg/mL, respectively (Figure 2 and Figure 3). The cytotoxicity (CC_50_), anti-HBV activity (IC_50_), and their corresponding therapeutic index (TI) values were also shown in Table 2.

### 2.4. Time-Dependent Downregulation of HBV Replication

Additionally, the time-dependent effects of both drugs on HBeAg production, a sign of active viral DNA replication, were studied. HBV DNA replication was suppressed in a time-dependent manner, consistent with anti-HBsAg action. Compounds **1** and **2** inhibited HBV replication by 26.7% and 38.2%, respectively, on day 3 post-treatment, whereas lamivudine, an antiretroviral drug, inhibited HBeAg production by 30.7% (Figure 4). Compounds **1** and **2** decreased HBV replication by 36.6% and 57.6%, respectively, on day 5 post-treatment, whereas lamivudine suppressed HBeAg formation by 59.5% (Figure 4). The experiment was halted on day 5 due to cell overgrowth. Remarkably, compound **2** showed a stronger inhibitory effect on HBV replication on day 3 post-treatment and was comparable on day 5 post-treatment to the currently available anti-HBV medicine, lamivudine (2.0 M).

## 3. Discussion

Worldwide, the hepatitis B virus (HBV) is a prevalent cause of hepatitis. Clinical trials have been conducted with various anti-HBV medications, including interferon and nucleotide analogs, but they have been associated with a number of side effects. As a result, the search for novel anti-HBV medications is of critical importance [20,21,22], and traditionally and naturally sourced novel anti-HBV medications that are efficient and affordable are a popular topic among those involved in the search for new drugs.

Assessing the expression of HBsAg as a measure for the detection of viral infection and HBeAg as a marker of viral DNA replication, the HBV-reporter cell line HepG2.2.15 remains a popular and globally used in vitro model for evaluating the anti-HBV agents [23,24]. Lamivudine or other NAs are used as reference drugs [25].

In a recent study, certain medicinal herbs from the Socotra Island were shown to possess anti-hepatitis B viral action. In particular, *D. cinnabari* showed promising anti-hepatitis B virus properties [3]. As part of our ongoing work on this medicinal plant, we isolated and elucidated the structure of two chalcone derivatives that are responsible for the exhibited anti-hepatitis B virus efficacy in vitro for the first time from the red resin of *D. cinnabari.*

Although the cell culture model reveals the reduction in HBV antigen expression following each treatment, it cannot distinguish whether the reduction is due to real antiviral activity against viral antigens or cytotoxicity. As a result, the isolated compounds were cytotoxically evaluated to identify a safe dose for anti-HBV investigations using the MTT assay.

Bioassay-guided fractionation of the methanolic extract of *D. cinnabari* and examination of the inhibitory effect on HBsAg expression led to the identification of 2,4’-dihydroxy-4-methoxydihydrochalcone (compound **1**) and 2,4’-dihydroxy-4-methoxyhydrochalcone (compound **2**), which were isolated for the first time from this interesting medicinal plant. Both compounds significantly inhibited the expression of HBsAg in a dose- and time-dependent way at noncytotoxic concentrations with IC_50_ values of 20.56 and 6.36 μg/mL, respectively. The isolated chalcones were also analyzed for time-dependent effects on HBeAg production, which is a marker of active viral DNA replication. Compared to the currently available anti-HBV medication (lamivudine), compound **2** demonstrated a stronger inhibitory effect on HBV replication at 3 days and an almost equivalent effect at 5 days after treatment.

Numerous studies have demonstrated that various types of chalcones can impact critical sites in viral-infection-related disorders. These potentially diversified antiviral bioactions render chalcone compounds an attractive broad-spectrum contender for combating the most recent viral pandemic (COVID-19) as well as any other developing viral infections [26]. Their intriguing biological properties have been investigated as potential pharmacological compounds having the ability to target a wide range of human viruses, such as hepatitis B virus (HBV), hepatitis C virus, MERS-CoV, SARS-CoV, human rhinovirus, herpes simplex, HIV, and influenza virus. Many viral molecular targets have been demonstrated to be affected by chalcones, including reverse transcriptase, protease, neuraminidase, aminotransferases, peroxide dismutase, glutathione peroxidase, and other related enzymes [27,28,29,30].

The role of chalcones in HBV has rarely been examined. Thiophenylindenone was shown to be the most effective in vitro inhibitor of DNA hybridization reported by Patil et al. [31], who synthesized many aryl/heteroaryl-substituted thienyl chalcones. Their findings showed that chalones may be interesting contenders for reducing DNA virus replication. Moreover, our results are in agreement with findings reported by Mathayan et al. [32], who tested the anti-HBV properties of *Pongamia pinnata* seeds containing certain chalcone derivatives. *P. pinnata* extract effectively suppressed HBV replication at a low dose with no evident cytotoxicity [32]. Additionally, molecular docking investigations revealed that two chalone derivatives isolated from *P. pinnata*, namely isopongachromene and glabaarachalcone, interact with the HBV DNA polymerase.

Furthermore, numerous chalcones isolated from *Glycyrrhiza uralensis* and *G. glabra* were evaluated for their anti-HCV activity [33]. Licochalcone A and isoliquiritigenin are natural examples that have an IC_50_ of 2.5 and 3.7 μg/mL, respectively, against HCV genotype 2a (J6/JFH1P47) [33]. Their efficacy was demonstrated by their ability to suppress the replication and production of HCV virus subgenomic RNA and protein synthesis [29]. Xanthohumol is another well-known chalcone for its antiviral properties that protect the liver [30]. Cells infected with HCV were treated with xanthohumol, which reduced aminotransferases, transforming growth factor b1 expression, and the hepatic steatosis score.

## 4. Materials and Methods

### 4.1. Plant Material Collection and Preparation

The resin of *D. cinnabari* was collected from the island of Socotra in Yemen in November 2014 and classified at the Pharmacognosy Department, Faculty of Pharmacy, Sana’a University. The specimen of assured quality was preserved at the Pharmacognosy Department, Faculty of Pharmacy, Sana’a University. Red resin (5 g) was macerated thrice in 300 mL methanol for 24 h, and the solvent was regularly filtered and evaporated using a rotary evaporator. Until usage, the dried methanolic extract was kept at −20 °C.

### 4.2. General Experimental Instruments

The ^1^H-, ^13^C-NMR, and 2D-NMR spectra were analyzed using a Bruker AMX-500 spectrometer and tetramethylsilane (TMS) (Karlsruhe, Germany) as an internal standard.

Chemical shifts (*δ*) are given in parts per million based on the tetramethylsilane internal standard. The mass was determined using a Jeol JMS-700 high-resolution mass spectrometer (Tokyo, Japan). The ionization energy was maintained at 70 e using the electron impact mode of ionization. The resolution was set to 10,000, and a direct probe was utilized with a temperature ramp setting of 50 °C at a rate of 32 °C per minute, with a maximum temperature of 350 °C. Thin-layer chromatography (TLC) was carried out on precoated silica gel F_254_ plates (E. Merck, Darmstadt, Germany) and column chromatography (CC) was performed on silica gel (100–200 or 200–300 mesh, Merck, Darmstadt, Germany). Detection was performed at 254 nm and by spraying with p-anisaldehyde/H_2_SO_4_ reagent.

### 4.3. Fractionation and Isolation of Active Compounds

The methanolic extract (3.0 g) of *D. cinnabari* was separated by column chromatography using a prepacked silica gel column (35 mm i.d. ×350 mm) to produce 13 fractions. The elution began with a gradient of dichloromethane: methanol (95:5), and the polarity was gradually increased to pure methanol. A total of 110 fractions (30–40 mL) were collected and pooled based on their TLC behavior to produce 13 fractions. Further purification of fraction 2 using a chromatotron (silica gel 60, 0.04–0.06 mm, and 1 mm), a mixture of chloroform and ethyl acetate (CHCl_3_: EtoAc, 9:1), and an increase in the proportion of EtoAc led to the isolation of compound **1** (90 mg). Fraction 11 was purified on an LH-20 column and methanol: water (9:1) (10 mm i.d. × 700 mm) to produce three subfractions (fraction 11a, 11b, and 11c). Fraction 11a required further purification on the chromatotron (silica gel 60, 0.04–0.06 mm, 1 mm) with a mixture of chloroform and methanol (CHCl_3_: MeOH, 99:1) and an increase in polarity according to the proportion of methanol to produce compound **2** (4 mg).

### 4.4. Cell Cultures and Drugs

HepG2.2.2.15 cells (generously provided by Dr. Shahid Jameel from the International Center for Genetic Engineering and Biotechnology, New Delhi, India) were grown in RPMI-1640 medium (Gibco, St. Louis, MO, USA) supplemented with 10% heat-inactivated bovine serum (Gibco, St. Louis, MO, USA), 1× penicillin–streptomycin and 1× sodium pyruvate streptomycin (HyClone Laboratories, Missouri, TX, USA) at 37 °C in a humid chamber with 5% CO_2_ supply. As a positive control, the drug lamivudine was utilized (Sigma, St. Louis, MO, USA).

### 4.5. Cytotoxicity Assessment

Using a 3-(4,5-dimethylthiazol-2-yl)-2,5-diphenyltetrazolium bromide (MTT) cell proliferation assay, the cytotoxicity of these compounds on HepG2.2.15 cells was examined to determine their concentrations without affecting the cell viability in preparation for future study. The cells were seeded in a flat-bottom 96-well tissue culture plate at a density of 1 × 10^4^ cells/100 μL/well (Corning Inc., Corning, NY, USA). At 24 h after incubation, cells were then treated (in triplicate) with various concentrations of the compounds (0, 6.25, 12.5, 25, 50, or 100 μg/mL) and cultured for an additional 48 h. There was no toxic effect because the final DMSO concentration in the assays was less than 0.1%. Additionally, the wells containing the medium only (blank) and the medium with 0.1% DMSO (untreated/negative control) were maintained under the culture conditions. MTT reagent (10 μL/well) was added to the cells, and they were then incubated for an additional 3–4 h. Each well was treated with 100 μL of detergent solution when their color turned purple, and then incubated for another hour. The optical density (OD) was measured at 570 nm using a microplate reader (BioRad, xMark, Hercules, CA, USA). A nonlinear regression test was carried out using Excel software to determine the CC_50_ value (the concentration causing 50% cytotoxicity) according to the following equation:(1)$$\mathrm{Survival}\;\mathrm{fraction}=\frac{OD[s]-OD\left[b\right]}{OD\left[c\right]-OD\left[b\right]}$$
where *OD*[*s*] is the absorbance of the sample, *OD*[*b*] is the absorbance of the blank, and *OD*[*c*] is the absorbance of the negative control. The cytotoxicity was also estimated from the number of dead cells by calculating the difference in the number of viable cells between untreated and treated groups, and is expressed as a percentage of total cell number.

### 4.6. Microscopic Examination

On days 1, 3, and 5 after the treatment, the cells were visually inspected to detect their morphological changes, such as cell membrane damage and cytoplasm uniformity, using an inverted microscope (Optica, Italy) at a magnification of 200×.

### 4.7. Dose- and Time-Dependent Analyses of HBsAg Expression in HepG2 Cells

HepG2 cells were seeded on 96-well plates at a density of 0.5 × 10^4^ cells per well and cultured at 37 °C. The following day, the culture media were replaced with fresh medium, (160 μL, each in triplicate) containing compounds at various concentrations (0–80 μg/mL), and incubated. The growth media were collected on day 3 and 5 after treatment and stored at −20 °C until use. The secreted HBsAg was determined using an enzyme immunoassay (ELISA) kit (Monolisa HBsAg ULTRA, BioRad, Hercules, CA, USA) according to the manufacturer’s instructions. The absorbance (OD) at 450 nm was determined using a microplate reader. Excel was used to calculate the IC_50_ values. The concentration (dose) of 50% inhibition was determined using nonlinear regression analysis (IC_50_).
(2)$$\mathrm{Inhibition}\;\%=\left(\frac{\;OD\left[C\right]-OD\left[T\right]}{OD\;\left[C\right)}\;\right)\times 100$$
where *OD*[*C*] and *OD*[*T*] indicate the absorbance of control and the test sample, respectively.

### 4.8. Analysis of the Time-Dependent Downregulation of HBV Replication

Based on the HBsAg inhibition results, the isolated compounds were subjected to a time-course (day 3 and 5) study on the downregulation of HBV replication by determining the HBeAg expression in the presence of 80 μg/mL of these compounds. The ELISA was performed using a HBeAg/Anti-HBe Elisa Kit (DIASource, Louvain-la-Neuve, Belgium).

### 4.9. Statistical Analysis

All data of triplicated samples are expressed as mean ± S.E.M. In a set of data, determination of total variation was performed by one-way analysis of variance (ANOVA), following the Dunnet’s-test (Excel 2010). *p* < 0.001 was considered significant.

## 5. Conclusions

In conclusion, the findings of this investigation revealed that *D. cinnabari* is a valuable source of potential medicinal chemicals for the treatment of hepatitis B. Two anti-hepatitis B chalcone derivatives, 2,4’-dihydroxy-4-methoxydihydrochalcone and 2,4’-dihydroxy-4-methoxyhydrochalcone, were isolated for the first time from this plant using bioassay-guided fractionation of the methanolic extract. These findings support the hypothesis that medicinal plants can be a good source of potential antiviral medicines. Based on the current findings, *D. cinnabari* will be chosen for future examination with the hope of discovering novel naturally occurring bioactive chemicals. Further research is required to verify the results and explain the mode of their antiviral actions.

## Figures and Tables

**Figure 1 molecules-27-00952-f001:**
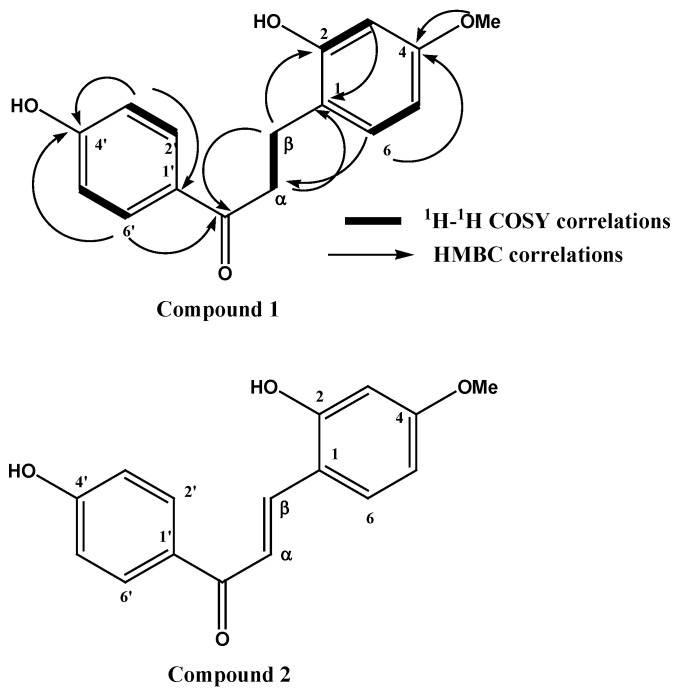
Chemical structure of the isolated chalcones.

**Figure 2 molecules-27-00952-f002:**
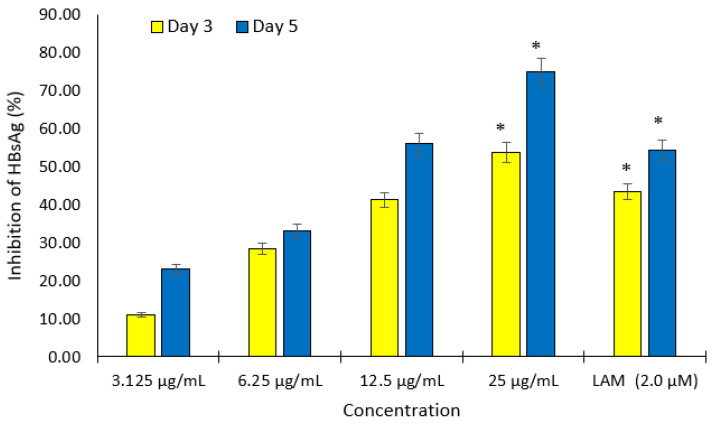
Dose- and time-dependent inhibitory effect of compound **1** on HBsAg expression in HepG2.2.15 culture. The antigen expression in the culture medium was measured using an ELISA to determine the anti-HBV activity of different concentrations of compound **1**. Lamivudine (2.0 µM) was used as the positive control. Data are presented as the mean ± standard error of the mean (*n* = 3), * *p* < 0.001 vs. positive control group.

**Figure 3 molecules-27-00952-f003:**
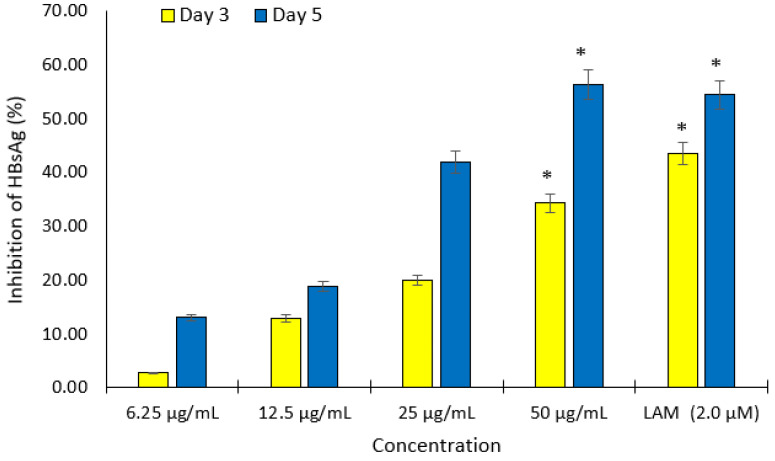
Dose- and time-dependent inhibitory effect of compound **2** on HBsAg expression in HepG2.2.15 culture. The antigen expression in the culture medium was measured using an ELISA to determine the anti-HBV activity of different concentrations of compound **2**. Lamivudine (2.0 µM) was used as the positive control. Data are presented as the mean ± standard error of the mean (*n* = 3), * *p* < 0.001 vs. positive control group.

**Figure 4 molecules-27-00952-f004:**
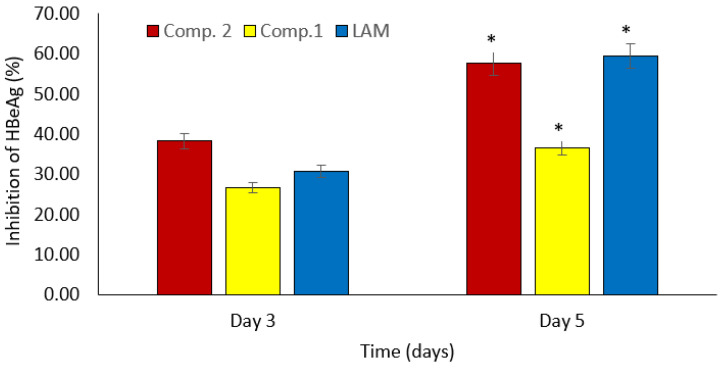
Effect of compounds **1** and **2** on time-dependent downregulation of HBeAg expression in HepG2.2.15 culture. The antigen expression in the culture medium was measured using an ELISA. Lamivudine (2.0 μM) was used as the positive control. Data are presented as the mean ± standard error of the mean, * *p* < 0.001, day 5 vs. day 3.

**Table 1 molecules-27-00952-t001:** ^1^H (500 MHz) and ^13^C NMR (125 MHz) spectral data of compounds **1** and **2** in CD_3_OD.

No	Comp. 1		Comp. 2	
	*δ*H	*δ*c	*δ*H	*δ*c
1	-	121.7	-	121.7
2	-	161.3	-	161.3
3	6.41 (1H, d, *J* = 2.3 Hz)	99.7	6.53 (1H, d, *J* = 2.0 Hz)	100.1
4	-	162.5	-	162.5
5	6.29 (1H, dd, *J* = 8.1, 2.3 Hz)	107.7	6.47 (1H, dd, *J* = 8.5, 2.0 Hz)	108.9
6	6.93 (1H, d, *J* = 8.1Hz)	131.3	7.60 (1H, d, *J* = 8.5 Hz)	133.8
α	3.1 (2H, t, *J* = 7.8 Hz)	40.0	7.42 (1H, d, *J* =15.7 Hz)	125.1
β	2.85 (2H, t, *J* = 7.8 Hz)	26.9	7.57 (1H, d, *J* =15.7 Hz)	144.2
1′	-	130.0	-	128.0
2′	7.87 (1H, d, *J* = 8.8 Hz)	131.9	7.51(1H, d, *J* = 8.6 Hz)	131.4
3′	6.83 (1H, d, *J* = 8.8 Hz)	116.2	6.83 (1H, d, *J* = 8.6 Hz)	116.9
4′	-	163.8	-	164.7
5′	6.83 (1H, d, *J* = 8.8 Hz)	116.2	6.83 (1H, d, *J* = 8.6 Hz)	116.9
6′	7.87 (1H, d, *J* = 8.8 Hz)	131.9	7.51(1H, d, *J* = 8.6 Hz)	131.4
4-OMe	3.78 (3H, s)	55.6	3.88 (3H, s)	56.1
C = O	-	201.8	-	193.1

**Table 2 molecules-27-00952-t002:** Results of cytotoxicity (CC_50_), anti-HBV activity (IC_50_), and their corresponding therapeutic index (TI) of compounds **1** and **2**. Values are the means of 3 determinations.

Samples	CC_50_ (μg/mL)	IC_50_ (μg/mL)	TI
Compound **1**	346.67	20.56	16.86
Compound **2**	242.11	6.36	38.08

## Data Availability

Data are available upon request.

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
