# Peer review of "Isolation and Characterization of Two Chalcone Derivatives with Anti-Hepatitis B Virus Activity from the Endemic Socotraen Dracaena cinnabari (Dragon’s Blood Tree)"

_molecules, 2022, doi:10.3390/molecules27030952_

Round 1

Reviewer 1 Report

To search a potentially effective natural medicine for hepatitis, the authors were continued to study the antiviral activities of local medicinal plants, and isolated two chalcone derivatives from Dracaena cinnabari (Dragon's blood tree) with anti-hepatitis activity. Besides, they also analyzed the chemical structures of these compounds, and then characterized their anti-HBV activities in vitro. The obtained results were strongly suggested that these chalcone derivatives might be effective, thereby proposing its possible usefulness as potential anti-hepatitis agents.

The experiments were well-designed and carefully carried out. Furthermore, the rationale of this study seemed reasonable and understandable, and the interpretation and presentation of obtained results seemed adequate and valid. The contribution of this work to medical sciences, particularly clinical treatment of viral hepatitis, seemed significant, and the findings presented here could be worth publishing.

However, the manuscript was considered to have many problems with its English, such as the applications of irrelevant prepositions, the misuse of a singular and plural form, the use of unnatural words and the unfavorable use of colloquial and slangy expression, Therefore, the drastic revision should be necessary to fix these problems. Of course, the assistance of native English speakers would be absolutely beneficial, thereby recommending strongly to obtain their help, if possible.

(Points should be revised)

  1. Lines 18-19: The phrase “cytotoxicity concentration”seemed insufficient a little, because the CC50 value is the cytotoxic concentration inducing the cell death by 50 %, and therefore it would probably be “50% cytotoxicity concentration”.
  2. Line 19: The word “resolved”seemed inappropriate, and it might be “determined” or else.
  3. Line 34: The phrase “NAs”was an abbreviated form of “nucleotide/nucleoside analogues”, and appeared at the first time in this line, but the abbreviation of this phrase was presented in lines 37-38, which was the second appearance. Generally, the abbreviation should be shown at the first appearance, namely in line 34.
  4. Line 34: The word “all”was not here, and would be moved to the beginning of sentence by replacing “The” in line 33.
  5. Line 51: The phrase “from diarrhea to dysentery to ulcers”seemed unnatural, and might be wrong, because of repeating a preposition “to” Perhaps, it might be “from diarrhea and dysentery to ulcers”.
  6. Line 56: The phrase “report now”would probably be better to say “now reported”.
  7. Line 67: The phrase “NMR data for compounds 1 and 2”might be good to say “NMR data of compound 1 and 2”. In this case, the word “compound” should be a singular form. Besides, the abbreviation “EIMS” was unidentified, but might possibly be guessed to denote “Electron Ionization Mass Spectrometry”. If so, the full spell of this method might be presented in front of its abbreviation.
  8. Line 68: The phrase “and corresponding with”might be better to denote “corresponding to” with putting a comma at the end of number “270”.
  9. Line 69: The phrase “assigned for”would probably be better to say “assigned to”.
  10. Lines 71-72: The phrase “those for”might be unnecessary, and should be deleted.
  11. Line 72: The word “missed”seemed not problem, but it would probably sound better to say “lacked” instead of “missed”.
  12. Lines 72-73: The phrase “with subsequent increase the number of methine by two”was awkward a little, and would probably be good enough to say “with increasing two methine”.
  13. Line 73: The phrase “1HNMR for 1”seemed unclear, and would be more evident to say “1HNMR for compound 1”.
  14. Line 75: The phrase “The remaining NMR protons signals assignedfor” would probably be necessary to be revised as “The remaining NMR proton signals were assigned to”.
  15. Line 83: The phrase “Collecting the above data”seemed awkward, and it might be better to say “The collecting data presented above” or “The accumulation of above data”.
  16. Line 85: The preposition “through”seemed strange a little, and would probably be better to be replaced with other word or phrase, for example “according to”.
  17. Line 88: The phrase “Spectral data for”might be better to say “Spectral data of”.
  18. Line 89: The phrase “those of 1”might be better to say exactly “those of compound 1”.
  19. Lines 90-92: This phrase seemed redundant and might be too long, and therefore could be quite difficult to understand distinctly what it intended to explain. As one possible measure of improving this phrase, it would be likely to revise this phrase. For example “Instead, two trans olefinic protons emerged and resonated atd52 (1H, J= 15.7 Hz) and (7.57 1H, J= 15.7 Hz), and they might be connected directly through one bond length to carbons at d 125.1 and 144.2 ppm in HSQC, and also assigned for position a and b, respectively”. Moreover, there were two points required to be confirmed. One was whether the preposition “through” instead of the word “though” (line 91) might be correctly used, and the other was that the word “carbos” might be “carbons”.
  20. Line 93: Similarly to the comment 17, “those of 1”should be “those of compound 1”.
  21. Line 95: The verb “are”should be a past form “were”.
  22. Line 97: The preposition “for”would be better to replace with “of”.
  23. Line 98: The word “Effect”should be a plural form “Effects”.
  24. Lines 99-100: The meaning of sentence “The cytotoxicity (CC50) of compounds 1 and 2 were evaluated to determine the appropriate concentration without cytotoxicity”seemed quite difficult, or rather might be impossible to understand. Since the CC50 value was one index to indicate the concentration of drug and chemicals to induce the 50% cytotoxicity, it would be a serious question how the CC50 value could be determined using the non-toxic concentrations of compound 1 and 2. In common-sense terms, the CC50 value could be calculated based on the correlation between the cytotoxic effect and the concentration of chemicals. Therefore, it would probably be impossible to determine the CC50 values of compounds in their non-toxic concentration range. So, it seemed possible to consider that this sentence might be wrong, and also might be reasonable to presume that this sentence would probably be “The cytotoxicities of compound 1 and 2 could be evaluated by determining the CC50 in their appropriate ranges of their toxic concentrations.
  25. Line 101; The phrase “no cytotoxicity”seemed too much to say, and it might be better to say “extremely week cytotoxicity”
  26. Line 103: The phrase “posttreatment with”seemed weird, and it would probably be better to say “after treatment with”.
  27. Line 105: The phrase “antiviral studies”was a slangy expression, and it might be better to say “studies of their antiviral effects”.
  28. Line 106: The abbreviation “HBsAg”should be described in a parenthesis following after the full length description “hepatitis B virus surface antigen”. Therefore, the subheading could be modified to be “3 Dose and time dependent inhibition of hepatitis B virus surface antigen (HBsAg) expression.
  29. Line 108: The phrase “at day 3 and 5 post-treatment”would probably be better to say “at day 3 and 5 after treatment”.
  30. Lines 110-111: The sentence “The HBsAg production was inhibited IC50 values 21.15,56 and 6.36 mg/ml of compound 1 and 2 respectively”seemed weird, and the structure of this sentence seemed incomplete, or rather might be wrong. Beside, the next sentence would similarly describe their inhibitory effects, just like repeating. Then, it would probably be good enough to say “The IC50 values of compound 1 and 2 could be estimated as 21.15, 20.56 and 6.36 mg/ml, respectively”.
  31. Line 113: The phrase “were shown also”seemed quite awkward, and it might be better to say “were also shown”.
  32. Line 116: The phrase “Time-course”should be “Time-dependent.
  33. Line 117: The phrase “the time course effect”should be “the time-dependent effects”.
  34. Line 118: The phrase “was studied”should be “were studied”.
  35. Line 121: The drug “lamivudine”might be necessary to explain more details, and it might be good enough to say “an antiretroviral drug lamivudine”.
  36. Line 124: The verb “had”seemed no problem, but it would probably be better to replace it with “showed”.
  37. Lines 130-132: As the title of Figure 2, the sentence “Dose and time dependent anti-HBV activities measured by ELISA showing inhibitions ofHBsAg expression in HepG2.2.15 culture supernatants by different concentrations of compound 1 and Lamivudine (2.0 mM) used as positive control”seemed quite redundant and awkward, and might be confusing. It would be better to simplify as follows; “Dose and time dependent inhibitory effect of compound 1 on HBsAg expression in HepG2.2.15 culture”, and then followed by the sentence “the antigen expression in the culture medium was measured using an ELISA to determine the anti-HBV activity of different concentrations of compound 1. Next, the sentence “Lamivudine (2.0 mM) was used as positive control”would be followed.
  38. Lines 134-136: The Figure 3 legend should be similarly revised as described in Figure 2, but “compound 1”should be replaced with “compound 2”.
  39. Line 138: The phrase “time-course”should be “time-dependent”.
  40. Line 139: The phrase “supernatants at days 3 and 5 measured byELISA” would probably be replaced with “the antigen expression in the culture medium was measured by an ELISA”.
  41. Lines 139-140: The sentence “Lamivudine (2.0 mM) usedas reference anti-HBV drug” seemed defective, and it should be “Lamivudine (2.0 mM) was used for reference of anti-HBV drug”.
  42. Line 144: The conjunction “but”instead of “however” might be better in this case.
  43. Lines 146-147: The structure of “finding high-efficiency, low-cost, and novel anti-HBV medications derived from traditional and natural sources is a popular topic in the hunt for new drugs”seemed unnatural and awkward, and this sentence would probably be better to say “the finding of novel anti-HBV medications with high-efficiency and low-cost from traditional and natural sources is a popular topic in the hunt for new drugs”.
  44. Lines 148-149: This long phrase “By assessing the expression of HBV surface antigen (HBsAg), a measureof viral infection, and HBeAg, a marker of viral DNA active replication” seemed quite weird and redundant, and might have difficulty to understand. Based on a mere conjecture, it would probably be possible to guess that this phrase might be “By assessing the expression of HBV surface antigen (HBsAg) as a measure for the detection of viral infection and a marker of active viral DNA replication”.
  45. Line 151: The phrase “the use of lamivudinein” seemed a little bit awkward, because of repeating “used” and “use”. Then, it would probably be better to say “the clinical application of lamivudine to”.
  46. Line 155 and 157: The abbreviation “HBV”seemed not truly necessary, and might be possible to delete them.
  47. Line 159: The phrase “Due to the fact”seemed weird, or rather might be wrong. It should be say “Despite the fact” or “In spite of the fact”. Moreover, the word “measures” seemed not fit here, and it would probably be better to replace with “reveals”.
  48. Lines 164-169: This was too long as one sentence, and therefore seemed quite difficult to understand the content of description. Therefore, it would probably be better to divide it into two parts before and after “and”in line 168. Of course, the conjunction “and” might not be necessary and should be deleted.
  49. Line 170: The phrase “for time-course effect”probably made this sentence awkward, and it might be good to say “to show its time-dependent effect” or “to elucidate its time-dependent effect”.
  50. Line 172: A comma attached to the end of “medication” was unnecessary, or rather might be confusing, and it should be removed.
  51. Lines 172-173: The phrase “a higher inhibitory effect on HBVreplication on day 3 after treatment and was equivalent on day 5 after therapy”seemed awkward, and it might be better to say “a higher inhibitory effect on HBV replication at 3 days and an almost equivalent effect at 5 days after treatment”.
  52. Line 179: The phrase “a range of”might be better to replace with “a wide range of”.
  53. Line 185: The phrase “in only”should be “only in”.
  54. Line 186: The preposition “by”seemed not good enough, and it would probably be better to say “reported by”.
  55. Lines 187-188: The phrase “many substituted aryl/heteroaryl derivedthienyl chalcones” seemed unnatural and awkward. It would be possible to guess that this phrase might be “many  aryl/heteroaryl-substituted thienyl chalcones”.
  56. Line 190: The word “reported”instead of “revealed” might be much better.
  57. Line 191: The phrase “seeds, which contain certain chalcone derivatives”was redundant, and it might be better to say “seeds containing certain chalcone derivatives”.
  58. Line 192: The phrase “with no evident of cytotoxicity”seemed unnatural, or rather might be wrong. It would probably be better to say either “with no evident cytotoxicity” or “without any evident cytotoxicity”.
  59. Line 194: The phrase “ pinnatanamely, isopongachromene and glabaarachalcone” had a strange structure. The comma was wrongly positioned, and therefore should be moved to the end of the plant name “P. pinnata”. Moreover, the phrase “interact to” was wrong, and should be “interact with”.
  60. Line 194: The word “protein”seemed unnecessary, and should be deleted.
  61. Lines 196-203: It seemed still question whether this part might be necessary or might be suitable to include in this place of Discussion.
  62. Line 208: The phrase “The specimen of assured quality”instead of “Voucher specimen” seemed appropriate and good.
  63. Line 209: The phrase “macerated trice with”seemed weird and impossible to understand. It would be possible to guess that this phrase might be “macerated thrice (three times) in”.
  64. Line 210: The phrase “for 24 hours with regular filtering and evaporating using a rotary evaporator”seemed extremely weird, and might have a quite odd structure. It would probably be necessary to entirely revise as follows. This phrase might be “for 24 hours, and the solvent was regularly filtered and evaporated using a rotary evaporator”.
  65. Line 213: The preposition “on”should be replaced with “using”.
  66. Line 214: The preposition “with”should be replaced with “and”.
  67. Line 215: The meaning of phrase “relative to”seemed not clear, and it would probably better to say “based on”.
  68. Line 217: The phrase “by using”should be either “by” or “using”.
  69. Line 225: The preposition “on”should be “by”.
  70. Line 226: The preposition “on”should be replaced with “using”.
  71. Line 229: The phrase “according to”would probably be better to replace with “based on”.
  72. Line 230: The preposition “on”should be replaced with “using”, and another preposition “with” should be replaced with “and”.
  73. Line 231: The phrase “and increasing the polarity by increasingthe proportion of EtoAc,” seemed complicate and hard to understand. It would probably be easier and better to say “with increasing the proportion of EtoAc”.
  74. Line 233: The verb “afford”seemed weird, and it might be better to say “give”.
  75. Line 235: The preposition “with”should be “and”.
  76. Line 236: The phrase “and increasing the polarity by”should be replaced with “with”.
  77. Line 239: The sentence “To grow HepG2.2.2.15 cells, RPMI-1640 medium (Gibco, USA) was used”seemed awkward, and it would probably be good to say “HepG2.2.2.15 cells were grown in RPMI-1640 medium (Gibco, USA) supplemented with”.
  78. Line 241: The commas attached at the end of “streptomy-cin”and “37oC” should be removed.
  79. Lines 246-248: The sentence “compounds cytotoxicity was tested on HepG2.2.15 cells to determinethe compound concentrations that did not affect cell viability and were used in subsequent tests.” seemed quite redundant and awkward, and might be necessary to revise totally. It would probably be better to say “the cytotoxicities of these compounds on 2.15 was examined to determine their concentrations without affecting the cell viability in preparation for future study”.
  80. Lines 248-249: The sentence “In 96-well tissue culture plates with flat bottoms, cells were seeded with 1x104cells/100 ml/well” seemed weird, and it would probably be better to say “The cells were seeded on a flat-bottom 96-well tissue culture plate at a density of 1x104 cells/100 ml/well”.
  81. Line 249: The phrase “After 24 hours of incubation, cells”would probably be revised to “At 24 hours after incubation, the cells”.
  82. Line 250: The phrase “with different compound concentrations”seemed weird, and it might be better to say “with different concentrations of the compounds”.
  83. Line 251: The sentence “There were no toxic effects”should be “There was no toxic effect,”
  84. Lines 253-254: The sentence “controls of blank (just media) and untreated/negative (0.1% DMSO in media) wereincluded” seemed quite weird and absolutely necessary to revise. It would probably be good to say “The wells containing the medium only (blank) and the medium with 0.1% DMSO (untreated/negative control) were maintained under the culture conditions”.
  85. Lines 256-257: The sentence “In a microplate reader, theoptical density (OD) was measured at 570 nm” seemed awkward, and it might be good say “The optical density (OD) was measured at 570 nm using a microplate reader”.
  86. Lines 257-259: The sentence “The following equation inExcel software was used in a non-linear regression test to determine the concentration that results in 50% cytotoxicity (CC50):” seemed totally disordered, in other word, just chaotic. It would be much better to say “A non-linear regression test was carried out using an Excel software to determine the CC50 value (the concentration causing 50% cytotoxicity) according to the following equation”.
  87. Lines 261-262: The sentence “Cytotoxicity was also measured as a percentage of cell viability relative to an untreated negative control”seemed awkward,or rather even wrong. It would probably be better to say “The cytotoxicity was also estimated from the number of dead cells by calculating the difference in the number of viable cells between untreated and treated groups, and then expressed as a percentage of total cell number”.
  88. Lines 264-267: It might be a big question whether the subsection 4.6. “Microscopic examination”might be necessary or not. No data obtained from this morphological study was actually not presented in this manuscript, expect a brief description in lines 102-104. Therefore, it seemed permissible to delete this part. In any case, the sentence “Cells were visually inspected at 1-, 3-, and 5-days post-treatment for morphological changes such as cell membrane defects and the uniformity of cytoplasmic components” seemed unnatural and awkward, it might be better to say “On 1-, 3-, and 5-days after the treatment, the cells were visually inspected to detect their morphological changes, such as the cell membrane damage and the cytoplasm uniformity ”.
  89. Line 268: The word “analysis”might be a plural form “analyses”.
  90. Line 269: The phrase “in 96 well”might be “on 96-well”.
  91. Line 271: The phrase “and controls”seemed unnecessary, and should be deleted.
  92. Lines 272-273: The sentence “The grown supernatant of each sample was harvested andstored at -20 ï½°C on day 3 and 5 following treatments” seemed weird and wrong, and it would probably be better to say “The growth media were collected on day 3 and 5 after treatments and stored at -20oC until use”.
  93. Line 274: The phrase “in the cultured supernatants”seemed unnecessary, and should be deleted.
  94. Lines 277-278: The sentence was extremely redundant, and therefore seemed good to replace the phrase “The concentration (dose) of 50% inhibition”with “The IC50 value”.
  95. Line 281:The structure of this subheading “8. Analysis of the time course of HBV replication downregulation” seemed quite weird, and it would probably be better to say “4.8. Analysis of the time course of HBV replication” or “4.8. Analysis of the time course of HBV downregulation”. Also, it might be possible, or rather probably much better to say “4.8. Analysis of the time-dependent downregulation of HBV replication”.
  96. Line 283: The phrase “of HBV replication downregulation”seemed unnatural, and it might be better to say “on the downregulation of HBV replication”.
  97. Lines 283-284: The phrase “using HBeAg expression at an 80 μg/ml concentration”seemed extremely unnatural and hard to understand these experiment conditions. It would probably good to say “by determining the HBeAg expression in the presence of 80 μg/ml of these compounds”.
  98. Line 284: The phrase “on culture media”seemed wrong and actually unnecessary. It should be deleted.
  99. Lines 284-285: The long phrase “according to the manufacturer's instructions using the HBeAg/Anti-HBe Elisa Kit (DIA-Source, Belgium)”seemed awkward, and it would be good to say “using the HBeAg/Anti-HBe Elisa Kit (DIA-Source, Belgium) according to the manufacturer's instructions”.
  100. Line 292: The phrase “for the first time from this plant”should be moved to the point between “isolated” and “using” in line 291.
  101. Line 292: The phrase “This study's findings”should be simply “These findings” or “The findings presented here”.
  102. Line 295: The phrase “More research”seemed unpopular, and it would be better to say “Further research” or “Further study”.
  103. Line 296: The phrase “the antiviral activity's mode of action”seemed unnatural and awkward, it would be better to say “the mode of their antiviral actions”.

Author Response

Reviewer 1

To search a potentially effective natural medicine for hepatitis, the authors were continued to study the antiviral activities of local medicinal plants, and isolated two chalcone derivatives from Dracaena cinnabari (Dragon's blood tree) with anti-hepatitis activity. Besides, they also analyzed the chemical structures of these compounds, and then characterized their anti-HBV activities in vitro. The obtained results were strongly suggested that these chalcone derivatives might be effective, thereby proposing its possible usefulness as potential anti-hepatitis agents.

The experiments were well-designed and carefully carried out. Furthermore, the rationale of this study seemed reasonable and understandable, and the interpretation and presentation of obtained results seemed adequate and valid. The contribution of this work to medical sciences, particularly clinical treatment of viral hepatitis, seemed significant, and the findings presented here could be worth publishing.

However, the manuscript was considered to have many problems with its English, such as the applications of irrelevant prepositions, the misuse of a singular and plural form, the use of unnatural words and the unfavorable use of colloquial and slangy expression, Therefore, the drastic revision should be necessary to fix these problems. Of course, the assistance of native English speakers would be absolutely beneficial, thereby recommending strongly to obtain their help, if possible.

Reply: We would like to express our gratitude to the respected reviewer for his or her assessment of the paper and for all insightful suggestions. Indeed, we gained a great deal of knowledge from all suggested corrections. One of the problems we faced was the high similarity (19%) especially in the section "Material and methods". We attempted to reduce the proportion, which necessitated rewriting and rephrasing paragraphs, resulting sometimes gin unnatural words and an unflattering use of colloquial and slangy expressions.

(Points should be revised)

  1. Lines 18-19: The phrase “cytotoxicity concentration”seemed insufficient a little, because the CC50 value is the cytotoxic concentration inducing the cell death by 50 %, and therefore it would probably be “50% cytotoxicity concentration”.

Reply: we are totally in agreement with. It is corrected in the revision.

  1. Line 19: The word “resolved”seemed inappropriate, and it might be “determined” or else.

Reply: It is corrected in the revision.

  1. Line 34: The phrase “NAs”was an abbreviated form of “nucleotide/nucleoside analogues”, and appeared at the first time in this line, but the abbreviation of this phrase was presented in lines 37-38, which was the second appearance. Generally, the abbreviation should be shown at the first appearance, namely in line 34.

Reply: we completely agree with. It is corrected in the revision.

  1. Line 34: The word “all”was not here, and would be moved to the beginning of sentence by replacing “The” in line 33.

Reply: It is replaced and corrected as suggested.

  1. Line 51: The phrase “from diarrhea to dysentery to ulcers”seemed unnatural, and might be wrong, because of repeating a preposition “to” Perhaps, it might be “from diarrhea and dysentery to ulcers”.

Reply: It is corrected as suggested.

  1. Line 56: The phrase “report now”would probably be better to say “now reported”.

Reply: It is corrected as suggested.

  1. Line 67: The phrase “NMR data for compounds 1 and 2”might be good to say “NMR data of compound 1 and 2”. In this case, the word “compound” should be a singular form. Besides, the abbreviation “EIMS” was unidentified, but might possibly be guessed to denote “Electron Ionization Mass Spectrometry”. If so, the full spell of this method might be presented in front of its abbreviation.

Reply: It is added and corrected in the revision.

  1. Line 68: The phrase “and corresponding with”might be better to denote “corresponding to” with putting a comma at the end of number “270”.

Reply: It is corrected as suggested.

  1. Line 69: The phrase “assigned for”would probably be better to say “assigned to”.
  2. Lines 71-72: The phrase “those for”might be unnecessary, and should be deleted.
  3. Line 72: The word “missed”seemed not problem, but it would probably sound better to say “lacked” instead of “missed”.
  4. Lines 72-73: The phrase “with subsequent increase the number of methine by two”was awkward a little, and would probably be good enough to say “with increasing two methine”.
  5. Line 73: The phrase “1HNMR for 1”seemed unclear, and would be more evident to say “1HNMR for compound 1”.
  6. Line 75: The phrase “The remaining NMR protons signals assignedfor” would probably be necessary to be revised as “The remaining NMR proton signals were assigned to”.
  7. Line 83: The phrase “Collecting the above data”seemed awkward, and it might be better to say “The collecting data presented above” or “The accumulation of above data”.
  8. Line 85: The preposition “through”seemed strange a little, and would probably be better to be replaced with other word or phrase, for example “according to”.
  9. Line 88: The phrase “Spectral data for”might be better to say “Spectral data of”.
  10. Line 89: The phrase “those of 1”might be better to say exactly “those of compound 1”.
  11. Lines 90-92: This phrase seemed redundant and might be too long, and therefore could be quite difficult to understand distinctly what it intended to explain. As one possible measure of improving this phrase, it would be likely to revise this phrase. For example “Instead, two trans olefinic protons emerged and resonated atd52 (1H, J= 15.7 Hz) and (7.57 1H, J= 15.7 Hz), and they might be connected directly through one bond length to carbons at d 125.1 and 144.2 ppm in HSQC, and also assigned for position a and b, respectively”. Moreover, there were two points required to be confirmed. One was whether the preposition “through” instead of the word “though” (line 91) might be correctly used, and the other was that the word “carbos” might be “carbons”.

Reply: The paragraph of the results of the structure elucidation using NMR was read again and corrected in many places as suggested. Please have a look to the revision.

  1. Line 93: Similarly to the comment 17, “those of 1”should be “those of compound 1”.

Reply: It is done suggested.

  1. Line 95: The verb “are”should be a past form “were”.

Reply: It is done as suggested.

  1. Line 97: The preposition “for”would be better to replace with “of”.

Reply: It is replaced as suggested.

  1. Line 98: The word “Effect”should be a plural form “Effects”.

Reply: It is done.

  1. Lines 99-100: The meaning of sentence “The cytotoxicity (CC50) of compounds 1 and 2 were evaluated to determine the appropriate concentration without cytotoxicity”seemed quite difficult, or rather might be impossible to understand. Since the CC50 value was one index to indicate the concentration of drug and chemicals to induce the 50% cytotoxicity, it would be a serious question how the CC50 value could be determined using the non-toxic concentrations of compound 1 and 2. In common-sense terms, the CC50 value could be calculated based on the correlation between the cytotoxic effect and the concentration of chemicals. Therefore, it would probably be impossible to determine the CC50 values of compounds in their non-toxic concentration range. So, it seemed possible to consider that this sentence might be wrong, and also might be reasonable to presume that this sentence would probably be “The cytotoxicities of compound 1 and 2 could be evaluated by determining the CC50 in their appropriate ranges of their toxic concentrations.

Reply: It was a rephrasing mistake. It is corrected to be clear for the readers.

  1. Line 101; The phrase “no cytotoxicity”seemed too much to say, and it might be better to say “extremely week cytotoxicity”

Reply: It is done as suggested.

  1. Line 103: The phrase “posttreatment with”seemed weird, and it would probably be better to say “after treatment with”.

Reply: It is corrected as suggested.

  1. Line 105: The phrase “antiviral studies”was a slangy expression, and it might be better to say “studies of their antiviral effects”.

Reply: It is done as suggested.

  1. Line 106: The abbreviation “HBsAg”should be described in a parenthesis following after the full length description “hepatitis B virus surface antigen”. Therefore, the subheading could be modified to be “3 Dose and time dependent inhibition of hepatitis B virus surface antigen (HBsAg) expression”.

Reply: We totally in agreement with. It is done as suggested.

  1. Line 108: The phrase “at day 3 and 5 post-treatment”would probably be better to say “at day 3 and 5 after treatment”.

Reply: It is done as suggested.

  1. Lines 110-111: The sentence “The HBsAg production was inhibited IC50 values 21.15,56 and 6.36 mg/ml of compound 1 and 2 respectively”seemed weird, and the structure of this sentence seemed incomplete, or rather might be wrong. Beside, the next sentence would similarly describe their inhibitory effects, just like repeating. Then, it would probably be good enough to say “The IC50 values of compound 1 and 2 could be estimated as 21.15, 20.56 and 6.36 mg/ml, respectively”.

Reply: Again it was a rephrasing mistake. It is corrected to be clear for the readers.

  1. Line 113: The phrase “were shown also”seemed quite awkward, and it might be better to say “were also shown”.

Reply: It is done as suggested.

  1. Line 116: The phrase “Time-course”should be “Time-dependent”.

Reply: It is corrected as suggested.

  1. Line 117: The phrase “the time course effect”should be “the time-dependent effects”.

Reply: It is done as suggested.

  1. Line 118: The phrase “was studied”should be “were studied”.

Reply: It is done as suggested.

  1. Line 121: The drug “lamivudine”might be necessary to explain more details, and it might be good enough to say “an antiretroviral drug lamivudine”.

Reply: It is added as suggested.

  1. Line 124: The verb “had”seemed no problem, but it would probably be better to replace it with “showed”.

Reply: we agree. It is replaced as suggested.

  1. Lines 130-132: As the title of Figure 2, the sentence “Dose and time dependent anti-HBV activities measured by ELISA showing inhibitions ofHBsAg expression in HepG2.2.15 culture supernatants by different concentrations of compound 1 and Lamivudine (2.0 mM) used as positive control”seemed quite redundant and awkward, and might be confusing. It would be better to simplify as follows; “Dose and time dependent inhibitory effect of compound 1 on HBsAg expression in HepG2.2.15 culture”, and then followed by the sentence “the antigen expression in the culture medium was measured using an ELISA to determine the anti-HBV activity of different concentrations of compound 1. Next, the sentence “Lamivudine (2.0 mM) was used as positive control”would be followed.
  2. Lines 134-136: The Figure 3 legend should be similarly revised as described in Figure 2, but “compound 1”should be replaced with “compound 2”.

Reply: The suggestion was taken in consideration. It is done in both figures as suggested.

  1. Line 138: The phrase “time-course”should be “time-dependent”.

Reply: It is corrected as suggested.

  1. Line 139: The phrase “supernatants at days 3 and 5 measured byELISA” would probably be replaced with “the antigen expression in the culture medium was measured by an ELISA”.

Reply: It is done as suggested.

  1. Lines 139-140: The sentence “Lamivudine (2.0 mM) usedas reference anti-HBV drug” seemed defective, and it should be “Lamivudine (2.0 mM) was used for reference of anti-HBV drug”.

Reply: It is corrected as suggested.

  1. Line 144: The conjunction “but”instead of “however” might be better in this case.
  2. Lines 146-147: The structure of “finding high-efficiency, low-cost, and novel anti-HBV medications derived from traditional and natural sources is a popular topic in the hunt for new drugs”seemed unnatural and awkward, and this sentence would probably be better to say “the finding of novel anti-HBV medications with high-efficiency and low-cost from traditional and natural sources is a popular topic in the hunt for new drugs”.

Reply: It is corrected as suggested.

  1. Lines 148-149: This long phrase “By assessing the expression of HBV surface antigen (HBsAg), a measureof viral infection, and HBeAg, a marker of viral DNA active replication” seemed quite weird and redundant, and might have difficulty to understand. Based on a mere conjecture, it would probably be possible to guess that this phrase might be “By assessing the expression of HBV surface antigen (HBsAg) as a measure for the detection of viral infection and a marker of active viral DNA replication”.
  1. Reply: It is changed as suggested.

  1. Line 151: The phrase “the use of lamivudinein” seemed a little bit awkward, because of repeating “used” and “use”. Then, it would probably be better to say “the clinical application of lamivudine to”.

Reply: It is corrected as suggested.

  1. Line 155 and 157: The abbreviation “HBV”seemed not truly necessary, and might be possible to delete them.

Reply: It is deleted as suggested.

  1. Line 159: The phrase “Due to the fact”seemed weird, or rather might be wrong. It should be say “Despite the fact” or “In spite of the fact”. Moreover, the word “measures” seemed not fit here, and it would probably be better to replace with “reveals”.

Reply: It is corrected as suggested.

  1. Lines 164-169: This was too long as one sentence, and therefore seemed quite difficult to understand the content of description. Therefore, it would probably be better to divide it into two parts before and after “and”in line 168. Of course, the conjunction “and” might not be necessary and should be deleted.

Reply: It is corrected as suggested.

  1. Line 170: The phrase “for time-course effect”probably made this sentence awkward, and it might be good to say “to show its time-dependent effect” or “to elucidate its time-dependent effect”.

Reply: It is corrected as suggested.

  1. Line 172: A comma attached to the end of “medication” was unnecessary, or rather might be confusing, and it should be removed.

Reply: It is removed as suggested.

  1. Lines 172-173: The phrase “a higher inhibitory effect on HBVreplication on day 3 after treatment and was equivalent on day 5 after therapy”seemed awkward, and it might be better to say “a higher inhibitory effect on HBV replication at 3 days and an almost equivalent effect at 5 days after treatment”.
  2. Line 179: The phrase “a range of”might be better to replace with “a wide range of”.

Reply: It is replaced as suggested.

  1. Line 185: The phrase “in only”should be “only in”.

Reply: It is corrected as suggested.

  1. Line 186: The preposition “by”seemed not good enough, and it would probably be better to say “reported by”.

Reply: It is corrected as suggested.

  1. Lines 187-188: The phrase “many substituted aryl/heteroaryl derivedthienyl chalcones” seemed unnatural and awkward. It would be possible to guess that this phrase might be “many  aryl/heteroaryl-substituted thienyl chalcones”.

Reply: It is corrected as suggested.

  1. Line 190: The word “reported”instead of “revealed” might be much better.

Reply: It is replaced as suggested.

  1. Line 191: The phrase “seeds, which contain certain chalcone derivatives”was redundant, and it might be better to say “seeds containing certain chalcone derivatives”.

Reply: It is corrected as suggested.

  1. Line 192: The phrase “with no evident of cytotoxicity”seemed unnatural, or rather might be wrong. It would probably be better to say either “with no evident cytotoxicity” or “without any evident cytotoxicity”.

Reply: It is corrected.

  1. Line 194: The phrase “ pinnatanamely, isopongachromene and glabaarachalcone” had a strange structure. The comma was wrongly positioned, and therefore should be moved to the end of the plant name “P. pinnata”. Moreover, the phrase “interact to” was wrong, and should be “interact with”.

Reply: It is corrected as suggested.

  1. Line 194: The word “protein”seemed unnecessary, and should be deleted.

Reply: It is deleted as suggested.

  1. Lines 196-203: It seemed still question whether this part might be necessary or might be suitable to include in this place of Discussion.

Reply: This paragraph, we believe, is also significant because it discusses other chalcone compounds that demonstrated antiviral action. As a result, we hope to leave it as it is.

  1. Line 208: The phrase “The specimen of assured quality”instead of “Voucher specimen” seemed appropriate and good.

Reply: It is changed as suggested.

  1. Line 209: The phrase “macerated trice with”seemed weird and impossible to understand. It would be possible to guess that this phrase might be “macerated thrice (three times) in”.

Reply: It is corrected as suggested.

  1. Line 210: The phrase “for 24 hours with regular filtering and evaporating using a rotary evaporator”seemed extremely weird, and might have a quite odd structure. It would probably be necessary to entirely revise as follows. This phrase might be “for 24 hours, and the solvent was regularly filtered and evaporated using a rotary evaporator”.

Reply: It is done as suggested.

  1. Line 213: The preposition “on”should be replaced with “using”.

Reply: It is replaced as suggested.

  1. Line 214: The preposition “with”should be replaced with “and”.

Reply: It is replaced as suggested.

  1. Line 215: The meaning of phrase “relative to”seemed not clear, and it would probably better to say “based on”.

Reply: It is corrected as suggested.

  1. Line 217: The phrase “by using”should be either “by” or “using”.

Reply: It is done.

  1. Line 225: The preposition “on”should be “by”.

Reply: It is done.

  1. Line 226: The preposition “on”should be replaced with “using”.

Reply: It is done.

  1. Line 229: The phrase “according to”would probably be better to replace with “based on”.

Reply: It is replaced.

  1. Line 230: The preposition “on”should be replaced with “using”, and another preposition “with” should be replaced with “and”.

Reply: It is done.

  1. Line 231: The phrase “and increasing the polarity by increasingthe proportion of EtoAc,” seemed complicate and hard to understand. It would probably be easier and better to say “with increasing the proportion of EtoAc”.

Reply: It is done.

  1. Line 233: The verb “afford”seemed weird, and it might be better to say “give”.

Reply: It is done.

  1. Line 235: The preposition “with”should be “and”.

Reply: It is done.

  1. Line 236: The phrase “and increasing the polarity by”should be replaced with “with”.

Reply: It is done.

  1. Line 239: The sentence “To grow HepG2.2.2.15 cells, RPMI-1640 medium (Gibco, USA) was used”seemed awkward, and it would probably be good to say “HepG2.2.2.15 cells were grown in RPMI-1640 medium (Gibco, USA) supplemented with”.

Reply: It is corrected.

  1. Line 241: The commas attached at the end of “streptomy-cin”and “37oC” should be removed.

Reply: It is removed.

  1. Lines 246-248: The sentence “compounds cytotoxicity was tested on HepG2.2.15 cells to determinethe compound concentrations that did not affect cell viability and were used in subsequent tests.” seemed quite redundant and awkward, and might be necessary to revise totally. It would probably be better to say “the cytotoxicities of these compounds on 2.15 was examined to determine their concentrations without affecting the cell viability in preparation for future study”.

Reply: it was a mistake by Rephrasing. It is corrected in the revision.

  1. Lines 248-249: The sentence “In 96-well tissue culture plates with flat bottoms, cells were seeded with 1x104cells/100 ml/well” seemed weird, and it would probably be better to say “The cells were seeded on a flat-bottom 96-well tissue culture plate at a density of 1x104 cells/100 ml/well”.

Reply: It is done.

  1. Line 249: The phrase “After 24 hours of incubation, cells”would probably be revised to “At 24 hours after incubation, the cells”.

Reply: It is done.

  1. Line 250: The phrase “with different compound concentrations”seemed weird, and it might be better to say “with different concentrations of the compounds”.

Reply: It is done.

  1. Line 251: The sentence “There were no toxic effects”should be “There was no toxic effect,”

Reply: It is corrected.

  1. Lines 253-254: The sentence “controls of blank (just media) and untreated/negative (0.1% DMSO in media) wereincluded” seemed quite weird and absolutely necessary to revise. It would probably be good to say “The wells containing the medium only (blank) and the medium with 0.1% DMSO (untreated/negative control) were maintained under the culture conditions”.

Reply: It is corrected.

  1. Lines 256-257: The sentence “In a microplate reader, theoptical density (OD) was measured at 570 nm” seemed awkward, and it might be good say “The optical density (OD) was measured at 570 nm using a microplate reader”.

Reply: It is changed in the revision.

  1. Lines 257-259: The sentence “The following equation inExcel software was used in a non-linear regression test to determine the concentration that results in 50% cytotoxicity (CC50):” seemed totally disordered, in other word, just chaotic. It would be much better to say “A non-linear regression test was carried out using an Excel software to determine the CC50 value (the concentration causing 50% cytotoxicity) according to the following equation”.

Reply: It is corrected.

  1. Lines 261-262: The sentence “Cytotoxicity was also measured as a percentage of cell viability relative to an untreated negative control”seemed awkward,or rather even wrong. It would probably be better to say “The cytotoxicity was also estimated from the number of dead cells by calculating the difference in the number of viable cells between untreated and treated groups, and then expressed as a percentage of total cell number”.

Reply: It is done.

  1. Lines 264-267: It might be a big question whether the subsection 4.6. “Microscopic examination”might be necessary or not. No data obtained from this morphological study was actually not presented in this manuscript, expect a brief description in lines 102-104. Therefore, it seemed permissible to delete this part. In any case, the sentence “Cells were visually inspected at 1-, 3-, and 5-days post-treatment for morphological changes such as cell membrane defects and the uniformity of cytoplasmic components” seemed unnatural and awkward, it might be better to say “On 1-, 3-, and 5-days after the treatment, the cells were visually inspected to detect their morphological changes, such as the cell membrane damage and the cytoplasm uniformity ”.

Reply: We think it is important to keep this part “Microscopic examination” since it was one of the experiments which was carried out. It is also corrected as suggested.

  1. Line 268: The word “analysis”might be a plural form “analyses”.

Reply: It is done.

  1. Line 269: The phrase “in 96 well”might be “on 96-well”.

Reply: It is done.

  1. Line 271: The phrase “and controls”seemed unnecessary, and should be deleted.

Reply: It is corrected.

  1. Lines 272-273: The sentence “The grown supernatant of each sample was harvested andstored at -20 ï½°C on day 3 and 5 following treatments” seemed weird and wrong, and it would probably be better to say “The growth media were collected on day 3 and 5 after treatments and stored at -20oC until use”.

Reply: It is done.

  1. Line 274: The phrase “in the cultured supernatants”seemed unnecessary, and should be deleted.

Reply: It is deleted.

  1. Lines 277-278: The sentence was extremely redundant, and therefore seemed good to replace the phrase “The concentration (dose) of 50% inhibition”with “The IC50 value”.

Reply: It is corrected as suggested.

  1. Line 281:The structure of this subheading “8. Analysis of the time course of HBV replication downregulation” seemed quite weird, and it would probably be better to say “4.8. Analysis of the time course of HBV replication” or “4.8. Analysis of the time course of HBV downregulation”. Also, it might be possible, or rather probably much better to say “4.8. Analysis of the time-dependent downregulation of HBV replication”.

Reply: The subheading is changed as suggested.

  1. Line 283: The phrase “of HBV replication downregulation”seemed unnatural, and it might be better to say “on the downregulation of HBV replication”.

Reply: It is done.

  1. Lines 283-284: The phrase “using HBeAg expression at an 80 μg/ml concentration”seemed extremely unnatural and hard to understand these experiment conditions. It would probably good to say “by determining the HBeAg expression in the presence of 80 μg/ml of these compounds”.

Reply: It is done.

  1. Line 284: The phrase “on culture media”seemed wrong and actually unnecessary. It should be deleted.

Reply: It is deleted.

  1. Lines 284-285: The long phrase “according to the manufacturer's instructions using the HBeAg/Anti-HBe Elisa Kit (DIA-Source, Belgium)”seemed awkward, and it would be good to say “using the HBeAg/Anti-HBe Elisa Kit (DIA-Source, Belgium) according to the manufacturer's instructions”.

Reply: It is corrected.

  1. Line 292: The phrase “for the first time from this plant”should be moved to the point between “isolated” and “using” in line 291.

Reply: It is done.

  1. Line 292: The phrase “This study's findings”should be simply “These findings” or “The findings presented here”.

Reply: It is done.

  1. Line 295: The phrase “More research”seemed unpopular, and it would be better to say “Further research” or “Further study”.

Reply: It is changed.

  1. Line 296: The phrase “the antiviral activity's mode of action”seemed unnatural and awkward, it would be better to say “the mode of their antiviral actions”. Reply: It is corrected.

Reviewer 2 Report

I admit the value of this work but this manuscript is too crude to be published in Molecules. Thus, my current decision is “Major revision”.

1.The author should provide basic data in supporting information including MS, 1H-, 13C-, DEPT, and 2D-NMR (COSY, HSQC and HMBC) spectra.

2.The author should provide purity tested compounds in materials and methods.

3.It does not make sense to provide duplicate data in Figures 2-4. The authors should merge Figures 2 and 3, then delete Figure 4.

4.The isolated compounds were identified using 2D spectroscopic data. Please provide the key correlations in figure.

5.Too many errors appear in this manuscript. For example, extra spaces or missing spaces always appear consecutively, and the font of compound number should be bold, “δ and J” should be italic, “7.87 (1H, d, J = 8.8 Hz” in table should be revised “7.87 (1H, d, J = 8.8 Hz)”. "CC50" should be revised to "CC50". The author should check and revise manuscript carefully. 

Author Response

Reviewer 2

I admit the value of this work but this manuscript is too crude to be published in Molecules. Thus, my current decision is “Major revision”.

Reply: We thank the respected reviewer for his opinion and we corrected a lot in the revision to put the manuscript in a better shape.

1.The author should provide basic data in supporting information including MS, 1H-, 13C-, DEPT, and 2D-NMR (COSY, HSQC and HMBC) spectra.

Reply: We attached the spectra of the isolated compounds as a supplementary file. The respected reviewer can have a look for all needed information.

2.The author should provide purity tested compounds in materials and methods.

Reply: The thin layer chromatography of both compounds was very clear and showed pure compounds. This is also to conclude from the NMR-spectra which showed clear signals indicating the purity of the isolated two compounds.

3.It does not make sense to provide duplicate data in Figures 2-4. The authors should merge Figures 2 and 3, then delete Figure 4.

Reply: The data are not duplicated. Figure 2 and 3 exhibited the dose and time dependent inhibitory effect of the isolated compounds on HBsAg expression in HepG2.2.15 culture whereas Figure 4 showed the effect of compounds 1 and 2 on time-dependent downregulation of HBeAg expression.

4.The isolated compounds were identified using 2D spectroscopic data. Please provide the key correlations in figure.

Reply: : because the isolated chemicals are well-known, we believe it is unnecessary to draw the correlations on the chemical structures. Typically, we do this with when the compounds are new. Nevertheless, it is accomplished during the revision as suggested.

5.Too many errors appear in this manuscript. For example, extra spaces or missing spaces always appear consecutively, and the font of compound number should be bold, “δ and J” should be italic, “7.87 (1H, d, = 8.8 Hz” in table should be revised “7.87 (1H, d, = 8.8 Hz)”. "CC50" should be revised to "CC50". The author should check and revise manuscript carefully. 

Reply: we checked the manuscript again and corrected and rephrased a lot in the revision. 

Reviewer 3 Report

The authors describe an interesting effect of two chalcone derivatives with anti-hepatitis B virus activity. However, there are some serious drawbacks of this manuscript that should be resolved.

  1. In the introduction section, there is no introduction to the main elements that are measured in the results section. For example, HBsAG, HBeAg, and Lamivudine are first mentioned in the Discussion section which makes reading the manuscript difficult.
  2. There is no description of why lamivudine was used as a positive control and how toxic this component is in comparison to compounds 1 and 2.
  3. What does it mean that: »This model was used to pioneer the use of lamivudine in the treatment of people infected with HBV.« The model described includes Hep cell line, not infected people.
  4. In section 4.5. Cytotoxicity assessment it is outlined: »…cytotoxicity was tested on HepG2.2.15 cells to determine the compound concentrations that did not affect cell viability…«. However, in the result section, there are no data showing how different concentrations of the compounds affect cell survival besides the CC50 that does not give information which concentration of the compounds is nontoxic for the Hep cells and if at that concentration the compound has an inhibitory effect on virus replication.
  5. There is no description of what a TI therapeutic index is and how it was calculated.
  6. There is no statistical evaluation of the significance of the results.

Minor comments:

  • In the introduction section it stays first “NAs” and then three lines later the NAs are described as nucleoside analogs.
  • In Figure 3. Lamivudine is abbreviated as LAM while in Figure 4 it is labeled as Lami.

Author Response

Reviewer 3

The authors describe an interesting effect of two chalcone derivatives with anti-hepatitis B virus activity. However, there are some serious drawbacks of this manuscript that should be resolved.

  1. In the introduction section, there is no introduction to the main elements that are measured in the results section. For example, HBsAG, HBeAg, and Lamivudine are first mentioned in the Discussion section which makes reading the manuscript difficult.

Reply: A statement on HBV encoded proteins (HBsAg and HBeAg) are now included in the Introduction section. Also, in an edited sentence, the NA-based drug lamivudine and others are mentioned.

  1. There is no description of why lamivudine was used as a positive control and how toxic this component is in comparison to compounds 1 and 2.

Reply: Lamivudine is the most popular and first generation FDA approved anti-HBV drug. This is nontoxic and widely used as reference drug in experimental settings, notably in cell culture models of HBV. Therefore, in this study we did not access its toxicity, and followed our previously optimized and published cell culture dose (2 mM).

References:

Parvez, M.K., Sehgal, D., Sarin, S.K., Basir, F.B., Jameel, S. 2006. Inhibition of hepatitis B virus DNA replication intermediate forms by recombinant IFN-g. World J. Gastroenterol. 12, 3006-3014.

Parvez, M.K, Ahmed, S., Al-Dosari, M.S., Abdelwahid, M.A.S., Arbab, A.H., Al-Rehaily, A.J., Al-Oqail, M.M. 2021. The novel anti-hepatitis B virus activity of Euphorbia schimperi and its quercetin and kaempferol derivatives. ACS Omega. doi:  10.1021/acsomega.1c04320.

Parvez, M.K., Al-Dosari, M.S., Arbab, A.H., Al-Rehaily, A.J., Abdelwahid, M.A.S. 2020. Bioassay-guided isolation of anti-hepatitis B virus flavonoid myricetin-3-O-rhamnoside along with quercetin from Guiera senegalensis leaves. Saudi Pharm. J. 28: 550-559.

Parvez, M.K., Rehman, M.T., Alam, P., Al-Dosari, M.S., Alqasoumi, S.I., Alajmi, M.F. 2019. Plant-derived antiviral drugs as novel hepatitis B virus inhibitors: cell culture and molecular docking study. Saudi Pharmaceutical Journal. 27:, 89-400.

Parvez, M.K., Al-Dosari, M.S., Alam, P., Rehman, M.T., Alajmi, M.F. 2019. The anti-hepatitis B virus therapeutic potential of anthraquinones derived from Aloe vera. Phytother. Res. 33, 1960-1970.

  1. What does it mean that: »This model was used to pioneer the use of lamivudine in the treatment of people infected with HBV.« The model described includes Hep cell line, not infected people.

 Reply: This statement is now rephrased with more clarity on the in vitro HepG2.2.15 culture model and the use of lamivudine as a reference drug.

  1. In section 4.5. Cytotoxicity assessment it is outlined: »…cytotoxicity was tested on HepG2.2.15 cells to determine the compound concentrations that did not affect cell viability…«. However, in the result section, there are no data showing how different concentrations of the compounds affect cell survival besides the CC50 that does not give information which concentration of the compounds is nontoxic for the Hep cells and if at that concentration the compound has an inhibitory effect on virus replication.

Reply: As per the globally accepted protocol, we tested our pure compounds up to the maximal dose of 50 mg/ml, which did not show any sign of cytototoxicity under visual observation (microscopic), too. Because this data is not important to show, we excluded this from the manuscript. However, the data of MTT based quantitative analysis was included. 

  1. There is no description of what a TI therapeutic index is and how it was calculated.

Reply: The therapeutic index is a quantitative measurement of the relative safety of a drug. It is a comparison of the amount of a therapeutic agent that causes the therapeutic effect to the amount that causes toxicity. It is calculated as:

CC50/IC50. If the respected reviewer allows, we think that it is not necessary to add this information to the table.

  1. There is no statistical evaluation of the significance of the results.

Reply: The missing statistical analysis section is now incorporated in Materials & methods section.

 Minor comments:

  • In the introduction section it stays first “NAs” and then three lines later the NAs are described as nucleoside analogs.

Reply: It is corrected in the revision.

  • In Figure 3. Lamivudine is abbreviated as LAM while in Figure 4 it is labeled as Lami.

Reply: it is corrected in the revision.

Round 2

Reviewer 1 Report

In this work, Dracaena cinnabari (Dragon's blood tree) was shown to have an anti-hepatitis activity derived from chalcone derivatives in this plant by analyzing their chemical structures and anti-HBV activities in vitro. The findings presented here were strongly suggested that these chalcone derivatives might be effective, thereby proposing its possible usefulness as potential anti-hepatitis agents.

The planning and performing of the experiments had no serious defect, and the interpretation and presentation of obtained results seemed adequate and valid. The contribution of this work to clinical treatment of viral hepatitis seemed significant, and the findings presented here could be worth publishing.

The manuscript might be totally revised, and English seemed good enough to report this work without any serious defect.

Author Response

Reviewer 1

In this work, Dracaena cinnabari (Dragon's blood tree) was shown to have an anti-hepatitis activity derived from chalcone derivatives in this plant by analyzing their chemical structures and anti-HBV activities in vitro. The findings presented here were strongly suggested that these chalcone derivatives might be effective, thereby proposing its possible usefulness as potential anti-hepatitis agents.

The planning and performing of the experiments had no serious defect, and the interpretation and presentation of obtained results seemed adequate and valid. The contribution of this work to clinical treatment of viral hepatitis seemed significant, and the findings presented here could be worth publishing.

The manuscript might be totally revised, and English seemed good enough to report this work without any serious defect.

Reply: We thank the respected reviewer for his opinion and all efforts and corrections done in the previous round.

Reviewer 2 Report

This paper has been revised and is recommended to be published in Molecules.

Author Response

Reviewer 2

This paper has been revised and is recommended to be published in Molecules.

Reply: We thank the respected reviewer for his opinion and all efforts and done in the previous round.

Reviewer 3 Report

The authors corrected the manuscript accordingly to most of the comments. However, the data a still not statistically evaluated which is a standard whenever some data are presented, Thus the data comparing the inhibitory effects of compounds 1, 2 and Lamivudine have to be statistically evaluated (Fig 2, 3, 4). 

Author Response

Reviewer 3

The authors corrected the manuscript accordingly to most of the comments. However, the data a still not statistically evaluated which is a standard whenever some data are presented, Thus the data comparing the inhibitory effects of compounds 1, 2 and Lamivudine have to be statistically evaluated (Fig 2, 3, 4). 

Reply: We thank the respected reviewer for the comment. In Fig 2 and Fig 3, the inhibitory effects of compound 1, and compound 2 were compared with Lamivudine (positive control). We amended the figures to show the statistical analysis. In Fig 4, the inhibitory effects of compound 1, compound 2, and Lamivudine were compared with their inhibitory effects on day 5 using ANOVA test. We amended the figure to show the statistical analysis.
